# Reliable Quantification of Ultratrace Selenium in Food, Beverages, and Water Samples by Cloud Point Extraction and Spectrometric Analysis

**DOI:** 10.3390/nu14173530

**Published:** 2022-08-27

**Authors:** Ingrid Hagarová, Lucia Nemček

**Affiliations:** Faculty of Natural Sciences, Institute of Laboratory Research on Geomaterials, Comenius University in Bratislava, 842 15 Bratislava, Slovakia

**Keywords:** cloud point extraction, trace selenium, spectrometric methods, beverages, food samples, water samples

## Abstract

Selenium is a trace element essential for the proper functioning of human body. Since it can only be obtained through our diet, knowing its concentrations in different food products is of particular importance. The measurement of selenium content in complex food matrices has traditionally been a challenge due to the very low concentrations involved. Some of the difficulties may arise from the abundance of various compounds, which are additionally present in examined material at different concentration levels. The solution to this problem is the efficient separation/preconcentration of selenium from the analyzed matrix, followed by its reliable quantification. This review offers an insight into cloud point extraction, a separation technique that is often used in conjunction with spectrometric analysis. The method allows for collecting information on selenium levels in waters of different complexity (drinking water, river and lake waters), beverages (wine, juices), and a broad range of food (cereals, legumes, fresh fruits and vegetables, tea, mushrooms, nuts, etc.).

## 1. Introduction

Selenium (Se) is an important micronutrient and a trace mineral essential for many living organisms, humans included. It has been systematically studied given its presence in many functionally active proteins. In 1973, Rotruck and his colleagues demonstrated that selenium is a key component of glutathione peroxidase (GSH-Px), which is an enzyme that inhibits lipid peroxidation in membranes [1]. To date, around 40 selenium-containing enzymes involved in a broad spectrum of biological activities have been identified [2], with GSH-Px being the most-studied one.

The importance of attaining adequate selenium intake throughout the life course to maintain health is indisputable; however, its elevated concentration in an organism or long-term supplementation can have adverse effects [3,4]. Toxic effects are mainly believed to be related to selenium’s ability to catalyze the oxidation of thiols and the generation of reactive oxygen species, which can damage cellular components [5]. The margin between selenium deficiency and toxicity is very narrow [6]. According to the World Health Organization (WHO), dietary deficiency occurs when daily intake of selenium falls below 40 μg/day, while exceeding an intake of 400 μg/day has been observed to be toxic to humans [7]. According to the available information, selenium toxicity is less common than problems associated with selenium deficiency [8]. It is estimated that almost one billion people are affected by selenium deficiency [9] and even more of the world’s population may have an inadequate selenium status.

Since selenium deficiency in humans has been associated with a number of chronic diseases, it is no surprise that many studies have been conducted to test various inorganic selenium compounds (including selenium nanoparticles) and organoselenium compounds for their effectiveness against certain types of cancer, for treatment of cardiovascular diseases, different types of viral and bacterial infections, neurodegenerative diseases, and more [2,10,11,12,13,14]. Food like nuts, cereals, oilseeds, seafood, red meat, and eggs are rich natural sources of selenium. However, the amount of selenium in food varies widely around the world, depending on the content of selenium in soils [15].

In natural systems, selenium exists in four oxidation states (−II, 0, +IV, and +VI). Inorganic species such as selenate (Se(VI)) and selenite (Se(IV)) predominate in water, whereas organic selenium compounds (methylated selenium, selenoaminoacids, and selenoproteins) are the major selenium species in foodstuffs and various biological matrices [16,17]. For instance, selenomethionine (SeMet) has been identified as the major selenocompound in cereal grains, grassland legumes, and soybeans, whereas methylselenocysteine (MeSeCys) is the major selenocompound in Se-enriched garlic, onions, sprouts, broccoli, and wild leeks [18]. For better illustration, some selenium species frequently found in environmental, biological, and food samples are depicted in Figure 1.

The fact that selenium is the focus of scientists who work in various sectors has been well-documented in many interesting review articles. Environmental analytical chemists have been analyzing an array of materials (mostly water, soil, and plants) to identify selenium species and concentration of this element in the environment. The goal is to deliver more information on how selenium affects various health conditions and diseases and on its involvement in biogeochemical cycles [19,20,21,22,23]. Many analytical chemists engaged in clinical analysis publish their research findings on total selenium content in various biological fluids and tissues [24,25,26,27] and on selenium species present in the analyzed materials [28]. The information on selenium concentration and forms in biological materials provide important data for biologists and other experts with deep expertise in medicine, who aim to explain the role of selenium in biological cycles and biochemical reactions that occur in the human body [29,30], as well as its role in supporting the body’s detoxification process [31]. This element has also attracted the interest of specialists in nutrition and dietetics who are interested in the development of the next-generation selenium supplements [32].

The analysis of various materials, including food products, is a complex process that requires a clear plan on how each stage should be addressed to obtain reliable results. It is critical to consider the entire experimental strategy from sample collection and storage to actual analysis. In the case of selenium quantification in food products, it is important to approach the analytical sample processing and the evaluation of the obtained results even more responsibly, since this type of material enters the human body directly and may affect selenium status in humans, which can have a positive or negative effect on our health. It is clear from the above that a reliable (accurate and precise) quantification of selenium in any material analyzed is the first step towards providing the answers to many questions that concern researchers in analytical science and related fields.

Selenium concentration in actual complex matrices such as food products is an important characteristic. Therefore, the reliable quantification of this element in real samples is of great significance. However, the determination of an ultratrace amount in a highly complex matrix is not an easy task. There are multiple detection methods that can be used for the quantification of selenium in different types of water samples, food, beverages, vegetable samples, fish, seafood, and meat samples. Spectrometric methods, such as UV-Vis spectrophotometry (UV-Vis), flame atomic absorption spectrometry (FAAS), electrothermal atomic absorption spectrometry (ETAAS), inductively coupled plasma optical emission spectrometry (ICP-OES), and inductively coupled plasma mass spectrometry (ICP-MS) are among the most common approaches. For the introduction of selenium-containing samples into spectrometric devices, hydride generation atomic absorption spectrometry (HG-AAS), hydride generation inductively coupled plasma optical emission spectrometry (HG-ICP-OES), hydride generation inductively coupled plasma mass spectrometry (HG-ICP-MS), and hydride generation atomic fluorescence spectrometry (HG-AFS) may be options, too. Notwithstanding, most of these methods are not sensitive enough for direct quantification of ultratrace selenium in actual complex matrices [33]. Even if using a highly sensitive method such as ICP-MS, which is currently considered the best practice among techniques allowing the trace and ultratrace analysis of elements [34,35,36,37], the presence of a co-existing matrix can be an issue when trying to achieve reliable results [38]. To be able to deal with these problems, an effective separation technique needs to be coupled with a reliable quantification method in order to deliver accurate results. Within a range of separation techniques, extractions appear to be the most popular approach [39]. With respect to the extraction phase, two main categories can be distinguished: liquid-based and solid-based extractions. Conventional techniques based on the liquid extraction principle include traditional liquid–liquid extraction (LLE), also known as solvent extraction. Since its introduction, the technique has undergone some significant changes; miniaturization is believed to be the most important of advancements. The application of small solvent volumes in LLE has led to the development of microextraction procedures such as liquid phase microextraction (LPME), solvent microextraction (SME), and liquid–liquid microextraction (LLME) [40]. Although the methods have been tailored slightly differently, they operate on the same principle of using small amounts of solvents for sample clean-up or target analyte extraction. There are three main separation modes of LPME: single drop microextraction (SDME), dispersive liquid–liquid microextraction (DLLME), and membrane-mediated liquid phase microextraction (such as hollow fiber liquid phase microextraction (HF-LPME) and solvent bar microextraction (SBME)) [41,42]. However, there are some other designs of the microextraction techniques mentioned above that have been reported [40,43]. 

In addition to miniaturization, a replacement of toxic or potentially toxic organic solvents with less harmful ones is another great improvement from the environmental point of view. Among green alternatives which can be effectively used for extraction, the nontoxic nonionic surfactants have found their niche in extraction procedures, specifically in cloud point extraction (CPE). This type of extraction has shown high efficiency in separation and preconcentration of (ultra)trace elements (including selenium) from various environmental, agricultural, biological, pharmaceutical, and food samples [44,45].

Reflecting the increased attention towards selenium concentrations in food, this research article is focused on papers addressing the reliable quantification of ultratrace selenium in liquid matrices such as water samples and beverages, as well as in solid food matrices. Papers published on CPE procedures used for the effective separation and preconcentration of (ultra)trace selenium before its reliable quantification by spectrometric methods have been reviewed in this article.

## 2. Cloud Point Extraction of Inorganic Ions

Cloud point extraction (CPE) is a surfactant-based extraction technique in which a nonionic surfactant acts as an extracting medium. A change in experimental conditions (mostly temperature, but also pressure, pH, and ionic strength) results in the formation of micelles and the aqueous surfactant solution separates into two isotropic phases: a phase rich in surfactant, which contains an analyte trapped by micellar structures (a surfactant-rich phase) and a bulk aqueous phase (a surfactant-poor phase or equilibrium solution).

The procedure for the separation and preconcentration of inorganic ions by CPE generally consists of several consecutive steps [46]. During step one, a complexing agent is added to the solution in order to form hydrophobic species of selected analyte, which remain in the hydrophobic core of the micellar structures in a surfactant-rich phase. This is followed by the addition of a surfactant with a concentration higher than the critical micellar concentration (CMC). Next, the solution is subjected to heating in a water bath (at a temperature higher than cloud point temperature (CPT)) to induce cloud point formation. At this stage, two isotropic phases are formed and their separation is usually accelerated by centrifugation. Subsequently, the system is cooled in order to enhance the viscosity of a surfactant-rich phase. The final step of CPE involves an aqueous phase removal. The highly viscous surfactant-rich phase is obtained and diluted by solution that is suitable for selected detector; for this purpose, mostly methanol or ethanol solution of a mineral acid is used. Such a diluted sample is now ready for quantification of a selected analyte. 

It is obvious that in order to develop a reliable procedure, some optimization of the experimental conditions needs to be done. There are two approaches to this issue. While the univariate strategy dominates the others due to an easier data interpretation, multivariate optimization strategies, which are faster, more economical, and more effective (fewer experiments need to be run), have been addressed in published papers as well [47,48].

In regard to CPE, the key components such as surfactants and complexing agents are seen as being able to contribute most effectively to the development of an effective procedure. While nonionic surfactant Triton X-114 (polyoxyethylene-7.5-octylphenoxy ether) seems to be the most popular extracting agent in CPE procedures (about 80% of studies in which CPE was employed for separation, preconcentration, and speciation of inorganic analytes utilized this surfactant) [45], when it comes to complexing agents, the range of products used is much wider (from less selective to highly selective).

For trace elements that can occur in a sample in multiple oxidation states, there are studies aimed at the development of a CPE procedure for selective separation of one particular oxidation state. If there are two inorganic forms of the element, such as Se(IV) and Se(VI), the optimization of experimental conditions is required to achieve the appropriate accuracy in selective separation and preconcentration of Se(IV). The total selenium content is determined after reduction of Se(VI) to Se(IV). Figure 2 illustrates the steps typically involved in selective separation and preconcentration of Se(IV) by CPE.

## 3. Different Approaches to Sample Preparation, Methodology, and Data Interpretation

The development of an accurate and effective CPE procedure requires considerable amount of time and effort. The optimization of experimental conditions is always carried out on model solutions. After validation of the proposed procedure by analyzing certified reference materials, the procedure is usually applied to the samples of interest. Sample pretreatment before the application of CPE procedure varies greatly. As a pretreatment step, natural water samples are usually filtered through a 0.45 µm pore-size membrane filter [49,50,51]. Filtration is preferably performed immediately after collection [50,51], and occasionally can be done just prior to use [47,52]. Tap water samples are often analyzed directly by CPE and do not require any special treatment [53]. Prior to analysis, beverages can simply be diluted, e.g., white wine samples [53], or digested by acids, e.g., pulp fruit juices [51]. Solid food needs to be decomposed before selenium quantification.

In a study on selenium content in green tea by Wen et al. [54], two portions of samples were prepared separately: water leachates for water-extractable selenium fraction, and decomposed samples for quantification of total selenium. The total Se content was approximately 3–5 µg/g, from which a water-extractable fraction was in the range of 25–35%. This value is quite high, although this may be primarily due to the unusual preparation of water-extractable fraction, which involved placing a beaker with tea samples into boiling water for 2 h. Even though there are many ways to prepare green tea, boiling the leaves in water for such a long time is definitely a rather rare approach. Thus, achieving such a high selenium intake through green tea consumption seems very unlikely.

There have been several CPE procedures proposed for separation and preconcentration of Se(IV). The aqueous matrices, in which inorganic selenium species predominate, require very little pretreatment with no influence on the oxidation state. Therefore, using the term ‘speciation analysis’ when referring to an analytical activity of identifying and measuring species is appropriate here. When dealing with water samples, a portion of the sample is analyzed for Se(IV) and a separate portion of the same sample is evaluated for total selenium content. Total selenium is always quantified after reduction of Se(VI) to Se(IV). Reduction with hydrochloric acid (HCl) is the most common approach [49,50,55], although reduction with L-cysteine [56] or with a mixture of L-cysteine and tartaric acid [51] have also been reported. 

The conversion of solid food into liquid always involves total decomposition of the sample, usually by means of mineral/oxidizing acids. As a result of oxidation, all selenium is present as Se(VI). After reduction of Se(VI) to Se(IV), a CPE procedure developed for separation and preconcentration of Se(IV) is applied, because all Se(VI) has been reduced to Se(IV). In this sense, Se(IV) actually represents the total selenium, which means that no speciation analysis was conducted. This is a possible source of confusion and misinterpretation of obtained data; some authors claim that they have determined the concentration of Se(IV), although in fact the total selenium content was measured. The above two paragraphs explain the difference between quantification of selenite concentration and quantification of total selenium. 

After sampling, solid food matrices are usually washed with deionized water, dried to a constant weight, ground into fine powder, and sieved [57]. After homogenization, an aliquot of the powdered sample is decomposed by acid digestion. In some cases, fresh samples are stored frozen or must be frozen before homogenization [51,58]. Acid decomposition by a mixture of nitric acid (HNO_3_) and hydrogen peroxide (H_2_O_2_) [51,53,54,58] or a mixture of HNO_3_ and perchloric acid (HClO_4_) [48,57] is typically followed by reduction of Se(VI) to Se(IV). Subsequently, an aliquot of such prepared solution is transferred to an extraction vessel and a CPE procedure for separation and preconcentration of Se(IV) is carried out.

An interesting sample of a hen egg was analyzed to gain information about selenium content [55]. Unfortunately, no information about pretreatment of the egg before its decomposition was provided in the published paper. The total selenium content in eggs was determined to be 43 µg/L. Some additional analyses were performed for water samples. While no selenium was quantified in tap water and drinking water samples, selenium content in river water samples was 71 µg/L. Concentration in the range of tens of µg/L in natural waters is considered as high. However, more detailed insight into river water sampling has not been given. Selenium level of 5 μg/L was imposed by the United States Environmental Protection Agency (USEPA) as a chronic aquatic life criterion [59,60]. It is the highest selenium concentration in surface water to which an aquatic community can be exposed indefinitely without resulting in an unacceptable effect [61]. Concentrations exceeding 5 μg/L pose a great hazard due to the possibility of selenium accumulation, e.g., in planktonic organisms [62]. In the light of this information, we can conclude that selenium concentration of 70 µg/L in water samples analyzed by Ulusoy [55] approaches the values reported for contaminated waters.

Rice, being a staple food in many countries worldwide, has been assessed thoroughly in order to obtain information about its main nutrients, vitamins, and minerals. In relation to selenium deficiency in countries where rice is the main dietary staple, Se-enriched rice could offer a solution to this problem. With regard to this issue, there are some studies worth mentioning. Sun et al. [57] performed several extraction procedures in order to distinguish between inorganic and organic selenium in Se-enriched rice samples. In the first step, aqueous leachates from Se-enriched rice flour were obtained. Then, cyclohexane was added to the liquid supernatant and the organic phase was separated from the aqueous phase. The latter was subjected to a CPE procedure and the amount of inorganic selenium was quantified. The total selenium content ranged from 124 to 143 ng/g and inorganic selenium represented only 14–24%. Selenium enrichment was achieved by spraying the rice leaves and seeds with nutrition agent containing organic selenium. Since the mean selenium concentration in rice is approximately 95 ng/g [63] and much lower values (14–85 ng per 1 g of rice) have been reported in Se-deficient regions [64], it can be concluded that total selenium levels in rice grains analyzed in this study were higher than average and that Se-enrichment procedure had the expected effect.

Tadayon and Mehrandoost [65] gathered samples from walnuts, measuring the amount of three selenium fractions by three different procedures. After preparation of a nut powder, Se(IV) was extracted from the powder by ultrasound-assisted extraction with deionized water as an extracting solution (i.e., selenite procedure). The determination of Se(VI) was carried out using ultrasound-assisted extraction with concentrated HCl followed by microwave digestion (i.e., selenate procedure), and digestion with concentrated HNO_3_ was required to leach selenomethionine into solution so that its content could be measured by UV-Vis spectrophotometry (i.e., selenomethionine procedure). Analytically, the latter method raises several questions. Is there any evidence that digestion with concentrated HNO_3_ results in the exclusive digestion of organic selenium? How do we know that organic selenium is solely in selenomethionine form? Furthermore, there is no information about total selenium concentrations in the paper. According to the results achieved, inorganic selenium accounted for only a negligible fraction, and selenomethionine concentration was within the range of 5.80–6.83 µg/g. 

Some answers to these questions are outlined in the paper by Güler et al. [66], in which they analyze commercially available vitamin tablets supplemented with selenium. The authors explain the terms ‘selenite procedure’, ‘selenate procedure’, and ‘organic selenium procedure’, and they also admit the presence of selenocysteine and some other organic selenium compounds besides selenomethionine. For this reason, researchers have termed digestion with concentrated HNO_3_ as ‘organic selenium procedure’ instead of ‘selenomethionine procedure’. Their hypothesis of more organic selenium compounds being present in vitamin tablets was confirmed by the results obtained during HPLC-ICP-MS measurements. For the purpose of this study, total selenium was quantified after digestion with concentrated HNO_3_ on a hot plate for 2 h. This procedure is similar to an ‘organic selenium procedure’. Eventually, the researchers correctly noted that even if they used the procedure proposed for organic selenium determination, the total selenium content was quantified. The selenate procedure allows for the extraction of both Se(IV) and Se(VI), so the total inorganic selenium content can be quantitatively estimated. The selenite procedure delivers information on Se(IV) content only, thus, the concentration of Se(VI) can be calculated as the difference between total inorganic Se and Se(IV). This explanation is fairly clear and one can agree with it. The labeling of the tablets played a crucial role in this study. If the label said the product contained inorganic selenium, researchers decided to use an extraction procedure for selenite and selenate. If selenium was present in an organic form, as specified by the manufacturer, extraction procedures for selenite, selenate, and organic selenium were utilized. In this work, samples of known composition were analyzed, the matrix could be considered uniform, and the speciation analysis results could be considered reliable.

For the purposes of selenium quantification in nuts, samples were crushed, and lipid fraction was separated from the non-lipid fraction [53]. The lipid fraction was removed by extraction with a mixture of methanol and chloroform, and the non-lipid fraction was dried and decomposed. While selenium content in sweet almonds, walnuts, and hazelnuts fell under the limit of detection (LOD) of CPE coupled to HG-ICP-OES, Brazil nuts were shown to contain approximately 51 µg/g of selenium in non-lipid fraction, which represented around 60% of the raw nut weight. If a daily selenium intake of 50–70 µg is considered optimal [7], consumption of one Brazil nut per day may be sufficient to overcome selenium deficiency.

For the simultaneous preconcentration and determination of selenium at trace levels in twelve food samples, Wang et al. [48] coupled a modified CPE procedure, so-called dual CPE (d-CPE), with hydride generation atomic fluorescence spectrometry (HG-AFS). Testing of seaweed, mushrooms, and grains for the presence of this element confirmed levels ranging from 35 to 99 ng/g. In laver, selenium content was as high as 182 ng/g. In d-CPE, the cloud point procedure is typically performed twice during the sample pretreatment process. At first, traditional CPE is applied and a surfactant-rich phase containing the analytes is obtained. During the second CPE procedure, the analytes are back-extracted into an aqueous phase [46]. A common drawback of coupling CPE with HG-AFS is a large volume of foam produced by surfactant during hydride generation. By d-CPE method, an aqueous extract is obtained, which can be injected directly into the spectrometer without any issues.

Using ultrasound in CPE procedures accelerates cloud formation. Altunay and Gürkan [58] used this type of energy and a nonionic surfactant, Brij 35, for separation and preconcentration of selenium in various fish and food samples, such as tea, mushrooms, fresh tomatoes, and cereals. Solid food samples were decomposed with a mixture of HNO_3_ and H_2_O_2_. After decomposition, the researchers expected selenium to be present in Se(VI) oxidation state. Decomposition was followed by reduction of Se(VI) to Se(IV) so that a CPE procedure proposed for separation and preconcentration of Se(IV) could be applied. We would like to stress here that since all Se(VI) has been reduced to Se(IV), the extracted Se(IV) actually represents total selenium in the sample. However, based on the information on selenium content in solid food given in this particular paper, the authors claim that Se(IV) levels were measured. On the other hand, when they repeated the same procedure on certified reference materials (CRMs), results indicated that total selenium content was determined, which is rather confusing. In terms of sample treatment prior to hydride generation atomic absorption spectrometry (HG-AAS) for selenium quantification, HCl is preferred over HNO_3_ due to the undesired formation of NO_x_ gases. Therefore, dilution of the surfactant-rich phase with 1M HNO_3_ rises a question. Why did Altunay and Gürkan [58] use HNO_3_ instead of HCl for their experiment? There is a broad range of researchers who use HCl for the dissolution of a surfactant-rich phase with satisfactory results. The authors themselves used 1M HCl in their previous work with no issues.

The thorough optimization of experimental conditions for ultrasound-assisted CPE (UA-CPE) with PONPE 7.5 as an extracting agent was carried out in order to obtain information about the selenium content in a broad range of samples [51]. The optimized UA-CPE procedure was applied to water samples of various complexity, pulp fruit juices, and solid food such as paddy rice, corn flour, fresh tomato, mushroom, and garlic. For solid food decomposition, the researchers used a mixture of HNO_3_ and H_2_O_2_, the same as in their follow-up studies. The additional information says that a mixture of 3M HNO_3_ was added to the samples separately from 2M H_2_O_2_ (3:2, *v*/*v*) so that selenium would not change its oxidation state from (IV) to (VI). Furthermore, the authors confusingly refer to Se(IV) content when using such type of decomposition for actual solid food samples and pulp fruit juices and the total selenium content when analyzing solid CRM.

An attempt to summarize the analytical characteristics reported in papers reviewed in this study resulted in the making of Table 1, which gives a detailed list of various spectrometric methods that have been utilized for quantification of selenium after its CPE separation. The optimization of experimental conditions has led to varying values of enhancement factors (EFs), ranging from 8 to 124, and varying values of limits of detection (LODs). While some combinations of methods (e.g., CPE and UV-Vis spectrophotometry [55,65]) enable quantification of trace selenium only, combination of CPE and ETAAS [49] or CPE and HG-AAS [51,58] allow for quantification of ultratrace selenium.

Several examples of analytical characteristics obtained during separation and preconcentration of Se(IV) in water and food products by means of some liquid microextraction procedures in conjunction with spectrometric detection can be seen in Table 2. The precision of procedures is expressed as relative standard deviation (RSD) and is comparable in all reviewed papers. It is rather rare to come across RSD values exceeding 5%.

The EFs varied considerably; for instance, EFs obtained by means of liquid microextraction procedures in conjunction with spectrometric detection were much higher compared to EFs obtained with CPE combined with a spectrometric technique (e.g., EF 500 [67]). Similarly, the LODs varied greatly for selenium determined by different combinations of methods. Regarding the effectiveness of particular combinations of liquid microextraction techniques and spectrometric methods, it has been shown that some combinations allow for reliable quantification of selenium in trace amounts only (for example, a combination of liquid microextractions with UV-Vis quantification [68,69]). Fortunately, finding a combination of methods allowing an accurate determination of selenium present in environmental samples at ultratrace concentration levels [33,67,70] is a relatively easy task.

**Table 2 nutrients-14-03530-t002:** Comparison of some analytical characteristics of several liquid microextraction methods proposed for Se(IV) separation and preconcentration and of methods for spectrometric quantification of this element.

Sample	Extraction	Detection System	Extraction Agent	EF	LOD (µg/L)	RSD (%)	Reference
Water, beverage, food	UA-IL-DLLME	ETAAS	[C_6_mim][Tf_2_N]	150	0.012	4.2	[71]
Water, beverage, food	MEA-IL-DLLME	ETAAS	[C_6_mim][PF_6_]	na	0.021	2.9	[72]
Water, garlic extract	on-line IL-DLLME	ETAAS	CYPHOS^®^ IL 101	20	0.015	5.1	[73]
Water	DLLME	ETAAS	CCl_4_	70	0.050	4.5	[74]
Water, food	SUPRAS-ME	ETAAS	1-octanol	na	0.10	4.3	[75]
Food	VA-IL-LLME	ETAAS	C_42_H_87_O_2_P	100	0.005	4.9	[70]
Edible oil	IL-DLLME	ETAAS	[C_12_mim][Tf_2_N]	140	0.03 *	5.1	[76]
Tea leaves, tea infusion	SFODME	ETV-ICP-MS	1-undecanol	500	0.00019	4.7	[67]
Water	DLLME	ETV-ICP-MS	CHCl_3_	64.8	0.047	7.2	[77]
Water	DLLME-SH-DES	UV-Vis	DES	315	0.76	5.5	[68]
Water, rice	IL-CI-AME	UV-Vis	[C_4_mim][PF_6_]	25	1.5	1.2	[69]
Water, beverage, food	VA-IL-DLLME	HG-AAS	[C_8_mim][Tf_2_N]	120	0.0015	2.7	[33]

* ng/g; EF: enhancement factor; LOD: limit of detection; RSD: relative standard deviation; na: not available; VA-IL-DLLME: vortex-assisted ionic liquid dispersive liquid–liquid microextraction; UA-IL-DLLME: ultrasound-assisted ionic liquid dispersive liquid–liquid microextraction; MEA-IL-DLLME: magnetic effervescent tablet-assisted ionic liquid dispersive liquid–liquid microextraction; IL-DLLME: ionic liquid dispersive liquid–liquid microextraction; DLLME: dispersive liquid–liquid microextraction; SFODME: solidified floating organic drop microextraction; SUPRAS-ME: supramolecular solvent microextraction; DLLME-SH-DES: dispersive liquid–liquid microextraction based on the solidification of a hydrophobic deep eutectic solvent; VA-IL-LLME: vortex-assisted ionic liquid liquid–liquid microextraction; IL-CI-AME: ionic liquid cold-induced aggregation microextraction; [C_8_mim][Tf_2_N]: 1-octyl-3-methylimidazolium bis(trifluoromethylsulfonyl)imide; [C_6_mim][Tf_2_N]: 1-hexyl-3-methylimidazolium bis(trifluoromethylsulfonyl)imide; [C_6_mim][PF_6_]: 1-hexyl-3-methylimidazolium hexafluorophosphate; CYPHOS^®^ IL 101: tetradecyl(trihexyl)phosphonium chloride IL; CCl_4_: carbon tetrachloride; CHCl_3_: chloroform; C_42_H_87_O_2_P: trihexyl(tetradecyl)phosphonium decanoate; [C_12_mim][Tf_2_N]: 1-dodecyl-3-methylimidazolium bis(trifluoromethylsulfonyl)imide; [C_4_mim][PF_6_]: 1-butyl-3-methylimidazolium hexafluorophosphate; DES: deep eutectic solvent.

## 4. Summary

The earliest use of the CPE separation technique is associated with procedures developed for trace element analysis in water matrices of various complexity. At that time, these procedures were very similar to each other and only differed in the trace element to be quantified and in the complexing agent used. Over a period of time, more complex matrices have been investigated and new modifications of established techniques have been proposed, among which a faster CPE (such as rapidly synergistic cloud point extraction (RS-CPE)) [52], a greener CPE (such as dual cloud point extraction (d-CPE)) [48], and a more sensitive CPE (such as mixed micelle cloud point extraction (MM-CPE)) [51] have been adopted for separation of selenium species from various complex food matrices.

The details about optimization of experimental conditions for CPE separation of Se(IV) were discussed in all research papers reviewed for the purpose of this study. In the examination of natural waters, where inorganic selenium species such as Se(IV) and Se(VI) appear to be predominant [78], some useful information on selenium species can be obtained by speciation analysis. A water sample is split into two sub-samples of known volume. In the first portion, Se(IV) is measured. In the second portion of the same water sample, total selenium concentration is determined after reduction of Se(VI) to Se(IV). However, the term ‘speciation analysis’ should not be used when referring solely to sample decomposition and determination of total selenium content. Our study also revealed that choosing the most appropriate decomposition method particularly matters if the material examined is solid food. There is a fair deal of confusion over sample decomposition and selenium oxidation state. Most of the papers summarized here utilize strong mineral acids to decompose the sample, which results in the oxidation of Se(IV) to Se(VI), i.e., in such decomposed samples, selenium is only present as Se(VI). 

Besides materials with natural levels of selenium, some Se-enriched products such as tea [54], hen eggs [55], and rice [57] have been subjected to analysis. The highest selenium content was reported in Brazil nuts (51 µg/g in non-lipid fraction) [53]. Although selenium content in Brazil nuts varies greatly, this nut species is believed to have the highest content of selenium among all foods [79] and cautious consumption of these nuts is recommended.

## 5. Conclusion

An increasing attention towards selenium can be attributed to its dual nature as an essential micronutrient that may become toxic when present at elevated levels. The narrow concentration range between selenium deficiency and toxicity contributes to the need for development of methods adapted for its reliable quantification at (ultra)trace concentration levels in complex matrices. The reliable quantification of (ultra)trace selenium in a complex matrix is often possible when an appropriate combination of an effective separation technique and quantification method is selected. Cloud point extraction is a separation technique that has found its way into (ultra)trace element analysis, including selenium.

In order for the CPE procedure to be effective, many experimental conditions need to be thoroughly optimized, which requires a lot of time. In this sense, multivariate optimization strategies which are less time-consuming, more economical, and more effective help accelerate the development of a reliable procedure. Although the optimization process is often accompanied by difficulties and challenges, thus developed CPE procedure offers high enhancement factors, quantitative recoveries, and acceptable precision. After validation, these procedures can be successfully used for separation and preconcentration of selected analytes from different samples. Some of the current procedures targeted at selenium were documented in the original research articles (published between 2006 and 2020) included and discussed in this overview.

## Figures and Tables

**Figure 1 nutrients-14-03530-f001:**
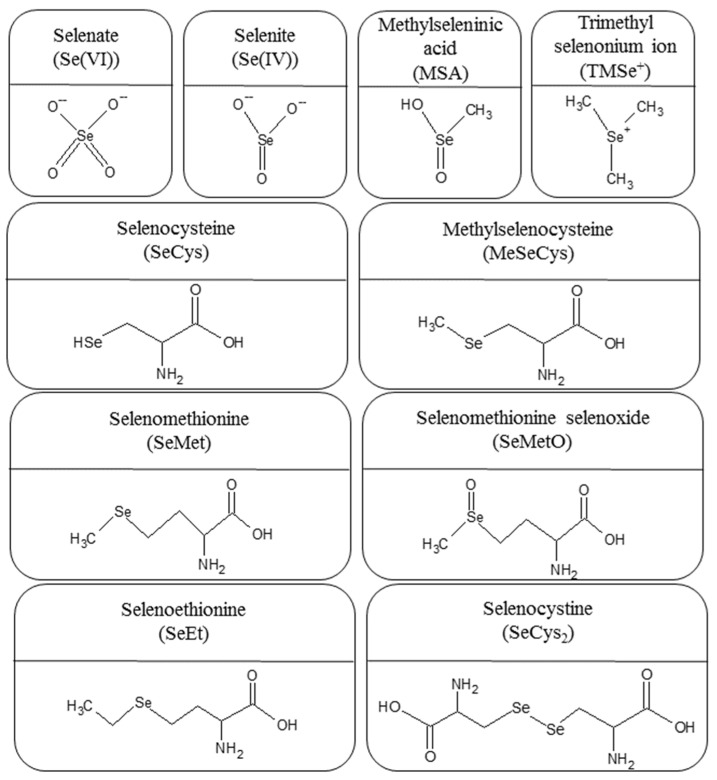
Some selenium species commonly detected in environmental, biological, and food samples.

**Figure 2 nutrients-14-03530-f002:**
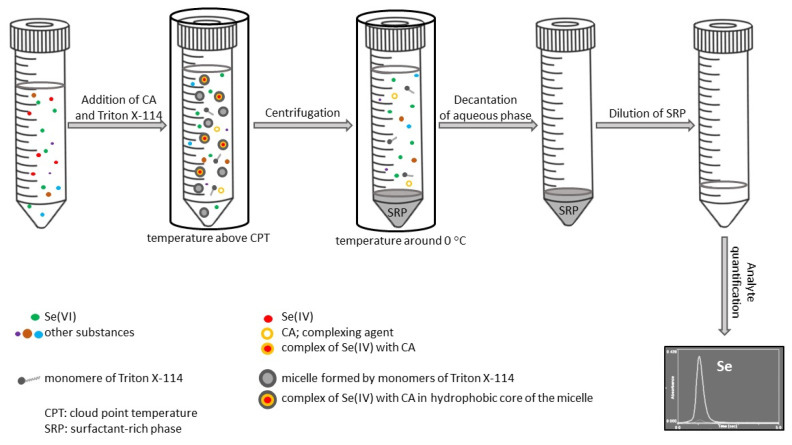
Schematic illustration of cloud point extraction steps for selective separation and preconcentration of Se(IV). Se, selenium.

**Table 1 nutrients-14-03530-t001:** Comparison of some analytical characteristics of the CPE methods proposed for Se(IV) separation and preconcentration and of methods for spectrometric quantification of this element.

Sample	Detection System	Complexing Reagent	Surfactant	Diluting Agent	EF	LOD (µg/L)	RSD (%)	Reference
Water, fish	ETAAS	DAB	TX-114	HNO_3_	100	0.0025	3.6	[49]
Water	ETAAS	OPD	TX-114	Met-OH	63.5	0.09	3.6	[47]
Rice	ETAAS	Dithizone	TX-114	Et-OH/HNO_3_	82	0.08	2.1	[57]
Tea	ICP-MS	DDTC	TX-100	HNO_3_	20	0.10	3.2	[54]
Water	ETV-ICP-MS	DDTC	TX-114	Et-OH	50	0.05	3.5	[56]
Water	ETV-ICP-MS	APDC	TX-114	Met-OH	39	0.008	3.9	[50]
Walnut	UV-Vis	DAN	TX-114	Met-OH	20	9.00	1.6	[65]
Water, hen egg	UV-Vis	DAHMP	TX-114/SDS	Et-OH	50	6.06	2.8	[55]
Water	UV-Vis	Dithizone	TX-114 (octanol) *	Met-OH	124	0.20	4.3	[52]
Water	UV-Vis	Dithizone	TX-114	Met-OH	103	0.30	3.2	[52]
Water, nuts, white wine	HG-ICP-OES	DDTP	TX-114	Met-OH/HCl	8	0.10	4.4	[53]
Various food	HG-AFS	APDC	TX-114	HCl + H_2_O_2_ **	11.8	0.023	4.0	[48]
Water, various food	HG-AAS	NRH^+^	PONPE 7.5/CTAB	HCl	155	0.00245	5.3	[51]
Fish, various food	HG-AAS	Toluidine Red	Brij 35	HNO_3_	140	0.0035	3.9	[58]
Vitamin supplements	Fluorometry	DAN	TX-114	H_2_O	10	2.10	5.0	[66]

* Rapidly synergistic CPE; ** dual CPE. EF: enhancement factor; LOD: limit of detection; RSD: relative standard deviation; ETAAS: electrothermal atomic absorption spectrometry; ICP-MS: inductively coupled plasma mass spectrometry; ETV-ICP-MS: electrothermal vaporization inductively coupled plasma mass spectrometry; UV-Vis: UV-Vis spectrophotometry; HG-ICP-OES: hydride generation inductively coupled plasma optical emission spectrometry; HG-AFS: hydride generation atomic fluorescence spectrometry; HG-AAS: hydride generation atomic absorption spectrometry; DAB: 3,3′-diaminobenzidine; OPD: o-phenylenediamine; DDTC: diethyldithiocarbamate; APDC: ammonium pyrrolidinedithiocarbamate; DAN: 2,3-diaminonaphthalene; DAHMP: 4,5-diamino-6-hydroxy-2-mercapto pyrimidine; DDTP: O,O-diethyl dithiophosphate; NRH+: neutral red, 3-amino-7-dimethylamino-2-methylphenazine hydrochloride; TX-114: Triton X-114; SDS: sodium dodecyl sulfate; CTAB: cetyltrimethylammonium bromide; HNO_3_: nitric acid; HCl: hydrochloric acid; H_2_O_2_: hydrogen peroxide; H_2_O: water; Met-OH: methanol; Et-OH: ethanol.

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
