# Peer review of "Reliable Quantification of Ultratrace Selenium in Food, Beverages, and Water Samples by Cloud Point Extraction and Spectrometric Analysis"

_nutrients, 2022, doi:10.3390/nu14173530_

Round 1

Reviewer 1 Report

Great and extensive review!!

I would suggest to include the ICP-MS in more details, It is considered a golden standard in this field.

English proofreading is however very necessary. 

Author Response

We thank you for your time spent carefully reviewing the manuscript. The corresponding changes and refinements made in the revised paper are summarized in our response below.

We apologize for not discussing the ICP-MS method in greater detail, it certainly deserves more attention. However, although ICP-MS method is the best practice for ultra-trace element analysis, we wanted to show the possibility to use also other, less sensitive spectrometric techniques as suitable quantification methods after an effective separation and preconcentration.

Without any doubt, the ICP-MS method has been constantly developing and improving; it is a topic worthy of consideration of writing a completely new manuscript on the basics of ICP-MS, its course of development and its applications in the field of environmental research.

Nonetheless, several contemporary references (with numbers 34-38) have been included in our manuscript which refer to ICP-MS used in ultratrace environmental analysis. In concerned papers, the method is discussed in more detail.

Our manuscript has been reviewed by a native English speaker. 

Reviewer 2 Report

The authors gave a short review for reliable determination of ultratrace selenium. The article is generally well written and very concise. I have several suggestion for authors:

1. Number of references is low for a review. Introduction must be expanded with, at least, dozen references

2. It should be thoroughly described why CPE was chosen and why is better of other similar techniques

3. I suggest adding a new table for comparing CPE with similar techniques (this will break long text and make it more attractive to readers)

Best regards

Author Response

We thank you for your time spent carefully reviewing the manuscript. The corresponding changes and refinements made in the revised paper are summarized in our response below.

Several new references have been added to the manuscript introduction, relevant to the topic: numbers 11-14 and numbers 19-32 refer to the literature on selenium; numbers 34-38 refer to the literature on ICP-MS method.

Research papers numbered 67-77 (references in Table 2) have also contributed to an increase in number of references in our manuscript.

The issue is quite complex and we do not dare to claim that CPE procedures are fast, simple and super effective for all types of analyzed matrices. We only want to point out that CPE is a type of extraction technique that can be used for separation and preconcentration of (ultra)trace Se(IV) in water as well as in a variety of food products. After optimizing the experimental parameters (this process is neither simple nor quick), it is possible to achieve high extraction yields, acceptable accuracy and a wide range of (often very high) preconcentration factors. CPE is only one of a wide variety of extraction techniques currently used in ultratrace analysis of inorganic analytes. A significant advantage of this method is that it requires only small volumes of non-toxic, non-volatile and inexpensive surfactants, which makes it a green alternative to conventional extraction techniques.

Table 2 was added to the manuscript. It shows the analytical characteristics obtained during separation and preconcentration of Se(IV) in water and food by means of some types of liquid microextraction procedures in conjunction with spectrometric detection.